# Radon Exposure and Cancer Risk: Assessing Genetic and Protein Markers in Affected Populations

**DOI:** 10.3390/biology14050506

**Published:** 2025-05-06

**Authors:** Yerlan Kashkinbayev, Baglan Kazhiyakhmetova, Nursulu Altaeva, Meirat Bakhtin, Pavel Tarlykov, Elena Saifulina, Moldir Aumalikova, Danara Ibrayeva, Aidos Bolatov

**Affiliations:** 1Institute of Radiobiology and Radiation Protection, Astana Medical University, Astana 010000, Kazakhstan; kashkinbaev.ye@gmail.com (Y.K.); bakhtin.m@amu.kz (M.B.); saifulina.e@amu.kz (E.S.); aumalikova.m@amu.kz (M.A.); danaraibrayeva@gmail.com (D.I.); 2Medical Genetics and Molecular Biology Department, Astana Medical University, Astana 010000, Kazakhstan; altaeva.n@amu.kz; 3National Center for the Biotechnology, Astana 010000, Kazakhstan; tarlykov@biocenter.kz; 4School of Medicine, Shenzhen University, Shenzhen 518060, China; bolatovaidos@gmail.com; 5School of Medicine, Astana Medical University, Astana 010000, Kazakhstan

**Keywords:** radon, lung cancer, protein markers, genes, epigenetic changes

## Abstract

Radon is a radioactive gas commonly found in the environment and is a major cause of lung cancer after smoking. It is invisible and accumulates indoors, particularly in poorly ventilated spaces. This review article summarizes current knowledge on genetic mutations and changes in protein expression linked to lung cancer caused by radon exposure. Recent research identified specific genetic mutations in tumor-related genes and alterations in proteins associated with inflammation, cell damage, and cancer development. Clarifying these molecular markers is essential for developing effective strategies for early detection, screening, and preventive measures against radon-induced lung cancer. These findings highlight the importance of regular radon monitoring, public health awareness, and targeted approaches to reduce the risk of radon-induced cancer. Advancing our understanding of the biological effects of radon will contribute to better outcomes for at-risk populations and inform future healthcare policies.

## 1. Introduction

Cancer remains a leading global cause of death, influenced by both environmental and genetic factors. Among these, radon, a naturally occurring radioactive gas, poses a significant but often underestimated health risk. Its widespread presence in residential and occupational settings, coupled with its established role as a leading cause of lung cancer after smoking, highlights the need for urgent research. Investigating the molecular mechanisms and biomarkers involved in radon-induced carcinogenesis is critical. Such studies could uncover genetic and protein-level changes caused by radon exposure, facilitating early detection, guiding public health interventions, and enabling personalized treatment approaches.

Radon is a naturally occurring radioactive gas produced by the decay of uranium found in soil, rock, and water. It is colorless, odorless, and tasteless, making it difficult to detect without specialized tools. Radon can accumulate in buildings, particularly in poorly ventilated spaces, and long-term exposure to elevated radon levels poses serious health risks, primarily increasing the likelihood of lung cancer [1,2,3]. The World Health Organization (WHO) estimates that radon is the second leading cause of lung cancer worldwide, following smoking. Research indicates that radon exposure is linked to approximately 21,000 lung cancer deaths each year in the United States alone [4]. The cancer-causing potential of radon stems mainly from the inhalation of radon decay products, which release alpha particles that can harm lung tissue and contribute to cancer development. Due to compelling epidemiological and experimental data linking radon exposure to lung cancer, researchers are increasingly focused on discovering genetic and protein indicators associated with radon-induced cancer development [5]. Understanding these biomarkers offers crucial insights into the molecular mechanisms behind tumor growth, potential risk contributors, and approaches for early detection and prevention. Alterations in crucial tumor suppressor genes, including TP53 and RASSF1A, along with oncogenes such as KRAS and EGFR, have been linked to radon-induced lung cancer development. Furthermore, proteins indicative of inflammation, oxidative stress, and DNA repair pathways play crucial roles in the carcinogenic effects stemming from radon exposure [6].

This review aims to comprehensively evaluate genetic and protein markers associated with radon-induced lung cancer, with a particular emphasis on populations living in areas with naturally high radiation levels. Although it is well-established that radon exposure significantly increases lung cancer risk, the exact genetic and molecular mechanisms involved remain incompletely understood. Currently, there is limited knowledge about specific biomarkers unique to radon-induced cancers and how these differ from cancers induced by other forms of radiation. Additionally, validated protocols for using these biomarkers clinically for early diagnosis or risk assessment are still lacking. By uncovering molecular changes caused by radon exposure, this review explicitly addresses these knowledge gaps by summarizing and critically analyzing both well-established and emerging genetic and protein biomarkers identified in recent genomic and proteomic studies. It examines the underlying molecular mechanisms and compares the genetic profiles of radon-induced cancers to those resulting from other radiation sources. Our synthesis clearly identifies promising but unvalidated markers and recommends future research directions essential for clinical translation. A key contribution of this work is the integration of multidisciplinary data to guide future directions in early diagnosis, personalized therapy, and targeted interventions for high-risk populations.

## 2. Background on Radon Exposure and Its Relevance to Public Health

Extensive epidemiological studies established a connection between radon exposure and lung cancer. A notable example is the European Pooling Study, which assessed data from multiple case–control studies and highlighted a clear dose–response relationship between residential radon levels and lung cancer risk [7]. The findings indicate that for each increase of 100 Bq/m^3^ in radon concentration, the lung cancer risk rises by approximately 16% [8]. Radon levels ca vary greatly based on geographic location, building design, and ventilation practices. Residences and workplaces situated in regions with a high concentration of natural uranium in the soil are particularly vulnerable to increased radon levels. The Environmental Protection Agency (EPA) advises that all homes below the third floor should be tested for radon, recommending remedial actions if levels exceed 148 Bq/m^3^ [9]. Mitigation strategies may include enhancing ventilation and sealing floors and walls to reduce radon entry.

The main sources of radon in indoor environments are the soil around the foundation, building materials, fuel, and household water. Radon concentration in residential areas depends not only on housing factors (such as type of housing, finishing materials, floor level, and year of construction), but also on environmental conditions (such as temperature, humidity, and atmospheric pressure), temporal factors (such as the season or day versus night), and ventilation performance for both indoor and outdoor air [10]. Recent studies increasingly highlighted the critical role of building materials, particularly concrete, in contributing to indoor radon accumulation. Radon, a naturally occurring radioactive gas, can emanate from construction materials containing uranium, thorium, and their decay products, leading to internal exposure among occupants. A comprehensive review by Bulut and Şahin (2024) underscored that the radon exhalation rate of concrete varies widely depending on factors such as the use of industrial by-products (e.g., fly ash, slag), porosity, moisture content, curing conditions, and the age of the material. These materials can contribute up to 25% of indoor radon levels, especially in poorly ventilated environments. Furthermore, their analysis indicated that specific additives (e.g., silica fume, rice husk ash) can modulate radon emissions, making material selection a potential mitigation strategy. This evolving body of research underscores the importance of integrating radon emission assessments into material safety evaluations and building codes, especially in radon-prone regions [11].

A cross-national review of radon regulations reveals considerable variation in how countries manage radon exposure in existing dwellings. According to IAEA data, Kazakhstan established a mandatory limit value of 200 Bq/m^3^, although it has not officially set a reference or action level. Several countries, including Bulgaria, Croatia, Cyprus, Montenegro, Romania, Serbia, Slovakia, Slovenia, and the Republic of North Macedonia, recommend a reference or action level of 400 Bq/m^3^, in line with European and WHO guidance. Latvia enforces one of the highest thresholds, with a reference level of 2000 Bq/m^3^ and a 600 Bq/m^3^ annual average action level. Other countries, such as Armenia and Belarus, regulate based on equilibrium equivalent concentrations (EEC) of radon progeny, using thresholds around 200 EEC. In contrast, Poland, Georgia, Estonia, and Greece currently lack defined reference, action, or limit values, indicating gaps in national radon policy frameworks. Slovenia takes a distinct approach by applying a dose-based threshold of 6 mSv/year. This regulatory inconsistency across countries highlights the need for globally harmonized radon standards to ensure effective risk mitigation and protect public health [12].

According to foreign sources, in southern regions of Spain and South Korea, living organisms are exposed to higher radon concentrations in new homes compared to old ones, and in renovated homes, the radon levels are higher than in unrenovated homes [13]. Experts from the International Commission on Radiological Protection (ICRP) state that radon, with an average volume activity of about 40 Bq/m^3^ in residential areas worldwide, is responsible for 10% of annually registered lung cancer cases (carcinoma, adenocarcinoma) [1]. Official data indicate that in certain regions of Kazakhstan, radon concentrations in soil reach up to 300,000 Bq/m^3^, and indoor levels range between 6000 and 12,000 Bq/m^3^, exceeding the permissible concentration limit by up to 60 times [14].

Effective public health policies are crucial for managing the risks associated with radon exposure. Numerous countries established regulations and guidelines for radon measurement and mitigation. As an example, the U.S. Environmental Protection Agency (EPA) manages the State and Tribal Indoor Radon Grants (SIRG) program, which allocates funds to support state and tribal radon initiatives aimed at reducing radon exposure. These grants enable state and tribal radon programs to implement and finance various radon risk reduction activities [15].

## 3. Overview of Radon-Induced Carcinogenesis

Radon-induced carcinogenesis primarily affects the lungs due to the inhalation of radon gas and its radioactive decay products, which emit ionizing radiation. This ionizing radiation, particularly alpha particles, interacts with lung tissue, causing cellular damage that can lead to cancer [16]. The carcinogenic process begins when radon gas is inhaled and decays into radioactive progeny, such as polonium-218 and polonium-214. These progenies emit alpha particles that have high linear energy transfer (LET), meaning they can cause dense ionization tracks within cells. When alpha particles collide with the DNA in lung epithelial cells, they induce complex DNA damage, including double-strand breaks and chromosomal aberrations [17]. The DNA damage caused by alpha particles can result in mutations if not properly repaired by the cell’s DNA repair mechanisms. These mutations can lead to uncontrolled cellular proliferation, a hallmark of cancer. Additionally, alpha particles can cause bystander effects, where non-irradiated cells exhibit damage signals due to intercellular communication with irradiated cells, further contributing to carcinogenesis [18]. Alpha particles have a limited range of penetration in biological tissues, approximately 40–90 μm, which is sufficient to affect the bronchial epithelium where lung cancer typically originates. The high LET radiation causes dense ionization tracks that are more likely to result in complex and irreparable DNA damage compared to low LET radiation, such as X-rays or gamma rays [19]. Radon-induced DNA damage can lead to genomic instability, characterized by an increased rate of mutations, chromosomal rearrangements, and aneuploidy. This genomic instability is a key factor in the initiation and progression of cancer. Over time, the accumulation of genetic alterations can activate oncogenes or inactivate tumor suppressor genes, driving the development of malignancies [20].

Epidemiological studies have consistently shown a link between radon exposure and lung cancer risk. Notably, studies of underground miners exposed to high levels of radon provided robust evidence of increased lung cancer incidence [21]. Animal studies also demonstrated the carcinogenic effects of radon, with increased tumor formation observed in rodents exposed to radon progeny [22]. The synergistic effects of radon and other risk factors, such as smoking, have been well documented. Smokers exposed to radon have a significantly higher risk of lung cancer compared to non-smokers, as tobacco smoke and radon progeny together produce a multiplicative effect of lung tissue damage and cancer risk [23].

International studies have shown that high radon concentrations in indoor environments increase the risk factor for developing lung cancer [24]. Previous model-based assessments estimated that radon exposure accounts for about 7% of lung cancer cases in Germany, 4% in the Netherlands, 20% in Sweden, 11% in Norway, and between 10% and 15% in the United States [25,26,27,28]. A recent global study covering 66 countries found that the median population-attributable risk (PAR) for lung cancer mortality due to residential radon exposure ranges from 13.6% to 16.5% [29]. These results are consistent with broader worldwide estimates indicating that radon exposure is responsible for roughly 10–20% of lung cancer cases and between 3% and 20% of deaths caused by lung cancer [30]. The United States Environmental Protection Agency (USEAP) reported that 13.4% of lung cancer deaths in 1995 were attributed to radon exposure [31]. Meanwhile, in the United Kingdom, over 1100 lung cancer deaths were linked to radon exposure in homes [32]. It is estimated that 1.2–1.9% of all cancer cases, and 1 in 10 lung cancer cases in Europe, may be caused by radon exposure indoors [33]. Thus, high radon concentrations in residential areas pose a potential danger to residents, primarily through inhalation of radioactive decay products, which can damage lung tissue and increase the risk of lung cancer [34].

## 4. Importance of Identifying Genetic and Protein Markers

Some of the biomarkers that have been studied in relation to radon-induced lung cancer include the following:DNA damage biomarkers: Radon exposure can cause DNA damage in lung cells, and specific biomarkers can indicate such damage. For example, measuring DNA adducts, (chemical modifications of DNA) or DNA strand breaks can provide insight into the extend of DNA damage caused by radon exposure [35].Genetic biomarkers: Certain genetic variations, including single nucleotide polymorphisms (SNPs), have been studied as potential biomarkers for increased susceptibility to radon-induced lung cancer. For instance, polymorphisms in DNA repair genes such as ERCC1 (rs3212986), XRCC1 (rs25487), and OGG1 (rs1052133) have been associated with heightened lung cancer risk among individuals exposed to radon. These genetic variations may influence DNA repair capabilities, inflammatory responses, and other processes involved in lung cancer development [36].Protein biomarkers: Proteins involved in various cellular processes, such as cell proliferation, apoptosis (programmed cell death), or inflammation, can serve as biomarkers for lung cancer. Researchers studied protein biomarkers in blood, sputum, or lung tissue samples to identify potential signs of radon-induced lung cancer [37].MicroRNA biomarkers: MicroRNA’s (miRNAs) are small RNA molecules that regulate gene expression. They also play a crucial role in various cellular processes, including cell development, proliferation, differentiation, growth control, and apoptosis. Studying miRNA expression profiles can help identify potential biomarkers for lung cancer caused by radon exposure [38,39].

Studies examined specific microRNAs that disrupt regulation in lung cancer and may be associated with radon exposure. Bersimbaev et al. (2020) and colleagues identified elevated levels of freely circulating miR-125b-5p and miR-155-5p in the plasma of lung cancer patients compared to a healthy control group. Therefore, these microRNAs play a key role in the pathogenesis of malignant lung neoplasms and may be considered as potential oncological markers for lung cancer diagnosis [5].

It is important to note that, although research in these areas continues, there are currently no validated biomarkers specific to radon-induced lung cancer that are widely used in clinical practice. The field of biomarkers research is complex, and further studies are needed to establish the clinical utility, sensitivity, and specificity of these biomarkers in radon-induced lung cancer.

Identifying genetic and protein markers in radon-induced cancer is crucial for several reasons. These markers play a pivotal role in understanding the mechanisms of carcinogenesis, improving early detection and diagnosis, and developing targeted therapies and personalized medicine approaches.

### 4.1. Understanding Carcinogenesis

Genetic and protein markers provide insights into the molecular mechanisms underlying radon-induced carcinogenesis. By identifying specific mutations and altered protein expressions, researchers can elucidate the pathways involved in the initiation and progression of cancer. For instance, studies have shown that radon exposure can lead to mutations in tumor suppressor genes such as TP53 and RB1, as well as oncogenes such as KRAS [40]. These genetic alterations disrupt normal cell cycle regulation and promote malignant transformation.

### 4.2. Early Detection and Diagnosis

Markers identified in radon-induced cancers can be used for early detection and diagnosis, which is critical for improving patient outcomes. Early detection allows for timely intervention and treatment, which can significantly enhance survival rates. For example, the presence of specific genetic mutations or elevated levels of particular proteins in bodily fluids (e.g., blood, sputum) can serve as biomarkers for early-stage lung cancer [41]. The development of non-invasive tests that detect these biomarkers can facilitate routine screening, particularly in populations at high risk of radon exposure.

### 4.3. Prognostic Value

Genetic and protein markers also have prognostic value, helping to predict the likely course and outcome of the disease. Certain markers are associated with more aggressive forms of cancer or a higher likelihood of recurrence. For example, overexpression of the protein epidermal growth factor receptor (EGFR) has been linked to poor prognosis in lung cancer patients [42]. Identifying such markers allows clinicians to stratify patients based on risk and tailor treatment strategies accordingly.

### 4.4. Targeted Therapies and Personalized Medicine

The identification of genetic and protein markers revolutionized the development of targeted therapies and personalized medicine. Targeted therapies are designed to specifically inhibit the molecular drivers of cancer, minimizing damage to normal cells and reducing side effects. For instance, drugs targeting activating mutations in the epidermal growth factor receptor (EGFR), such as gefitinib, have shown significant efficacy in patients with non-small cell lung cancer, particularly among those with radon-associated tumors harboring these alterations [43].

Beyond individual patient care, understanding genetic and protein markers in radon-induced cancer has broader public health implications. These markers can inform risk assessment and preventive strategies, particularly in populations living in high-radon areas. By identifying individuals with genetic predispositions to radon-induced cancer, targeted interventions and monitoring programs can be implemented to mitigate risk. Additionally, public health policies can be shaped by insights gained from molecular studies, leading to more effective radon mitigation efforts and awareness campaigns.

Despite significant progress, current research has not conclusively validated biomarkers specific enough to reliably distinguish radon-induced cancers from those caused by other radiation types. Additionally, the predictive value and clinical applicability of these biomarkers remain largely unknown. These gaps underscore the urgent need for standardized biomarker validation studies and larger epidemiological research efforts.

## 5. Genetic and Protein Markers in Radon-Induced Cancer

Identification and the role of specific genetic mutations. Radon-induced cancer, particularly lung cancer, has been associated with several specific genetic mutations. These mutations play a crucial role in the initiation and progression of cancer by disrupting normal cellular functions and promoting malignant transformation [44]. Understanding these genetic alterations is essential for developing targeted therapies and improving diagnostic and prognostic strategies.

Several studies identified key genetic mutations associated with radon-induced lung cancer. One of the most frequently mutated genes in this context is the tumor suppressor gene TP53. Mutations in TP53 are found in a significant proportion of radon-induced lung cancers and are associated with the loss of normal p53 protein function, which is critical for DNA repair, cell cycle regulation, and apoptosis [45]. The TP53 mutation in radon-induced cancers often involves transversions, a type of mutation indicative of exposure to ionizing radiation [46].

Another important gene implicated in radon-induced lung cancer is KRAS, an oncogene that, when mutated, leads to uncontrolled cell proliferation. Mutations in KRAS are found in a subset of radon-exposed individuals and are particularly prevalent in adenocarcinomas of the lung. The KRAS mutations often involve codon 12, which is a known hotspot for mutations in smoking-related and radiation-induced lung cancers [47].

In addition to TP53 and KRAS, mutations in the EGFR gene are also observed in some-radon induced lung cancers. These mutations typically occur in the tyrosine kinase domain, leading to constitutive activation of the EGFR signaling pathways, which promotes cell proliferation and survival [48]. EGFR mutations are particularly relevant for targeted therapies, as they can be effectively inhibited by drugs such as gefitinib and erlotinib.

Lim et al. (2019) noted that tumors with elevated radon levels are characterized by a significant presence of genes responsible for DNA damage and repair such as ATR, ATRX, BARD1, RAD50, SMARCA4, and TP53. Additionally, recent studies found that mutations in TP53 and promoter methylation of the RASSF1A gene occur more frequently in smokers with squamous cell lung cancer compared to non-smokers with adenocarcinoma [49].

Choi et al. (2018) studied lung tumors in non-smoking patients using NGS and found that mutations in CHD4, TSC2, and AR were more commonly observed in individuals exposed to high radon concentrations (>100 Bq/m^3^) [50].

Another gene of interest is anaplastic lymphoma kinase (ALK), which can undergo rearrangement and form fusion proteins that drive oncogenesis. ALK rearrangements are less common and have been identified in radon-induced lung cancers and are associated with sensitivity to ALK inhibitors, such crizotinib [51].

Moreover, some studies highlighted the role of the ROS1 gene, which, similar to ALK, can form oncogenic fusion proteins. ROS1 rearrangements respond to ROS1 inhibitors [52].

The mutations identified in radon-induced cancers disrupt critical cellular pathways, contributing to the development and progression of cancer. TP53 mutations result in the loss of tumor suppressor functions, allowing cells with damaged DNA to proliferate and accumulate further genetic abnormalities. This loss of function is particularly detrimental because p53 plays a vital role in responding to DNA damage induced by alpha particles from radon progeny [53]. Mutations in KRAS lead to the constitutive activation of the Ras protein, which is involved in signaling pathways that regulate cell growth and survival. The persistent activation of these pathways promotes oncogenesis by driving the proliferation of cells with genomic instability. This effect is compounded in the context of radon exposure, where the ionizing radiation from radon decay products causes extensive DNA damage [54]. EGFR mutations result in the constitutive activation of the EGFR protein, which promotes cell proliferation and survival through downstream signaling pathways, such as the PI3K/AKT and MAPK pathways. These mutations are significant because they can be targeted by specific inhibitors, providing a therapeutic avenue for treating radon-induced lung cancer [55]. ALK and ROS1 rearrangements create fusion proteins that drive oncogenesis by activating multiple signaling pathways involved in cells growth and survival [56]. These genetic alterations are important not only for their role in cancer development, but also for their therapeutic implications, as they can be targeted by specific inhibitors.

The identification of specific genetic mutations in radon-induced cancer has significant implications for diagnosis and treatment. Genetic testing for TP53, KRAS, EGFR, ALK, and ROS1 mutations can aid in the early detection of lung cancer in individuals exposed to radon. Furthermore, understanding the molecular profile of these tumors can inform treatment decisions, allowing for the use of targeted therapies that specifically inhibit the pathways activated by these mutations [57]. For instance, patients with EGFR mutations may benefit from EGFR inhibitors such as gefitinib or erlotinib, while those with KRAS mutations may require alternative strategies due to the limited efficacy of current KRAS-targeted therapies [58]. Patients with ALK or ROS1 rearrangements can be treated with ALK or ROS1 inhibitors, such as crizotinib, which have shown effectiveness in clinical trials [59].

## 6. Genomic Studies and Findings

Genomic studies significantly advanced our understanding of the genetic alterations associated with radon-induced cancer. These studies employ high-throughput sequencing technologies to identify somatic mutations, copy number variations, and other genomic alterations that drive carcinogenesis. Recent findings provided valuable insights into the molecular landscape of radon-induced lung cancer and highlighted potential targets for therapeutic intervention [60].

Comprehensive genomic profiling using next-generation sequencing (NGS) revealed a complex mutational landscape in radon-induced lung cancer. A study by Alexandrov et al. (2016) utilized whole-genome sequencing to analyze the mutational signatures associated with radon exposure. The study identified a unique mutational signature characterized by a high prevalence of C > T transitions, which is indicative of the DNA damage caused by alpha particles from radon progeny. This mutational signature provides a molecular fingerprint that can help distinguish radon-induced cancers from those caused by other etiological factors [61].

Recent genomic studies identified several driver mutations in radon-induced lung cancer. In addition to TP53, KRAS, EGFR, ALK, and ROS1, other genes such as BRAF, MET, and HER2 have been found to harbor mutations that contribute to oncogenesis. For instance, mutations in the BRAF gene, which encodes a serine/threonine kinase involved in the MAPK signaling pathway, have been detected in a subset of radon-induced lung cancers. The mutations, particularly the V600E mutation, result in constitutive kinase activity and are associated with increased tumorigenic potential [62].

Similarly, radon exposure has been linked to genetic alterations contributing to lung tumorigenesis, including changes in the MET proto-oncogene. Experimental models of radon-induced lung cancer in rats revealed frequent chromosomal losses in regions homologous to human 7q21–36, where the MET gene is located. These findings align with human lung cancer studies that report recurrent deletions or amplifications involving MET, a receptor tyrosine kinase that regulates cellular growth, survival, and motility [63].

Beyond the well-characterized mutations, emerging genomic alterations have been identified in radon-induced lung cancer. Whole-exome sequencing studies uncovered novel mutations and gene fusions that contribute to the disease. For example, recent research identified novel fusions involving the NTRK1 gene, which encodes a receptor tyrosine kinase involved in neuronal development. NTRK1 fusions lead to constitutive kinase activity and have been implicated in various cancers, including radon-induced lung cancer. These fusions represent potential targets for TRK inhibitors, which have shown efficacy in clinical trials [64].

Another important aspect of genomic studies in radon-induced cancer is the assessment of tumor mutational burden (TMB) and neoantigen load. High TMB is associated with increased neoantigen production, which can enhance immunogenicity of tumors and make them more responsive to immune checkpoint inhibitors. A study by Rizvi et al. (2018) demonstrated that radon-induced lung cancers often exhibit high TMB, suggesting that these tumors may benefit from immunotherapy. This finding underscores the potential for personalized immunotherapeutic approaches in treating radon-induced cancers [65].

Genomic studies also identified genetic alterations that correlate with response to therapy in radon-induced lung cancer. For example, the presence of EGFR mutations or ALK rearrangements is associated with sensitivity to tyrosine kinase inhibitors, while KRAS mutation often predicts resistance to these therapies. Understanding these genomic correlates is essential for optimizing treatment strategies and improving patient outcomes. A study by Skoulidis et al. (2018) highlighted the importance of genomic profiling in guiding therapy, demonstrating that patients with specific genetic alterations showed differential responses to targeted therapies and immunotherapies [66]. Table 1 summarizes the genetic markers previously studied in radon-induced lung cancer.

## 7. Comparison of Genetic Markers in Radon-Induced Cancer Versus Other Radiation-Induced Cancers

The genetic alterations observed in radon-induced cancer share similarities with those seen in other radiation-induced cancers, yet there are distinct differences influenced by the type and source of radiation. Comparing these genetic markers helps to elucidate the unique and overlapping pathways of carcinogenesis induced by different types of ionizing radiation [7].

Radiation-induced cancers, including those caused by radon, often exhibit mutations in key tumor suppressor genes and oncogenes. The most commonly affected genes across various radiation-induced cancers include TP53, KRAS, EGFR, and RB1. These genes are critical in regulating cell cycle, DNA repair, and apoptosis, making them primary targets of radiation-induced DNA damage. Certain genetic alterations are more specific to radon-induced cancer due to the unique properties of alpha radiation. For example, alpha particles cause dense ionization tracks, leading to complex DNA damage, which can result in specific types of mutations not as commonly seen with other forms of radiation [79] (Table 2).

Radiation exposure, whether from radon, gamma rays, or X-rays, leads to distinct mutational signatures and types of cancer. The unique properties of each type of radiation result in different patterns of DNA damage and subsequent genetic alterations. Thus, radon, a naturally occurring radioactive gas, primarily emits highly ionizing alpha particles. When radon and its progeny are inhaled, these alpha particles interact with the lung tissue, causing dense ionization tracks that lead to complex DNA damage. This interaction results in a high prevalence of transversions, specifically C > T and A > T mutations, along with intricate DNA rearrangements [83]. These mutational signatures are particularly indicative of the heavy and localized damage caused by alpha particles. The predominant type of cancer associated with radon exposure is lung cancer. The mutational profiles of lung cancer induced by radon exposure are distinct due to the unique DNA damage patterns caused by alpha radiation. These profiles help in identifying the cancer as radon-induced, setting it apart from lung cancers caused by other factors.

In contrast to alpha radiation, gamma rays and X-rays are forms of electromagnetic radiation with high energy but low ionizing capability compared to alpha particles. These types of radiation primarily cause single- and double-strand breaks in DNA, leading to a different set of genetic alterations. The mutational signature of gamma and X-ray radiation includes a higher prevalence of deletions, insertions, and point mutations [84]. This pattern of DNA damage is reflective of the more diffuse and penetrating nature of gamma and X-ray radiation, which affects a broader area of the genome. Gamma and X-ray radiation are associated with a variety of cancers, including leukemia, thyroid cancer, and secondary malignancies that arise after radiotherapy. The diversity in cancer types stems from the widespread nature of the DNA damage caused by these forms of radiation, which can affect multiple tissues and organs [85].

Understanding the similarities and differences in genetic markers between radon-induced and other radiation-induced cancers has important implications for research and clinical practice. It helps in the development of specific diagnostic tools, targeted therapies, and personalized medicine approaches tailored to the mutational profiles of different radiation-induced cancers.

## 8. Protein Markers in Radon-Induced Cancer

Understanding the role of protein markers in radon-induced cancer is critical for elucidating the biological mechanisms of radon carcinogenesis and identifying potential targets for therapeutic intervention. Key proteins involved in the response to radon exposure are primarily associated with DNA damage repair, cell cycle regulation, and apoptotic pathways. These proteins help maintain genomic stability and prevent malignant transformation in response to radon-induced DNA damage [86].

Several protein biomarkers have shown promising results in lung cancer diagnosis and monitoring. Here some examples:Carcinoembryonic antigen (CEA): CEA is a protein that often increases in lung cancer and is associated with tumor growth and metastasis. It can be measured in blood samples and used as a marker for lung cancer progression and treatment response [87].Cyfra 21-1: This protein biomarker, derived from cytokeratins 19 and 21, is elevated in certain types of lung cancer, particularly squamous cell carcinoma. It can be detected in blood samples and may assist in diagnosing and monitoring the disease [88].Progastrin-releasing peptide (ProGRP): ProGRP is a protein biomarker that may be elevated in small cell lung cancer (SCLC). It is used to aid in the diagnosis and monitoring of SCLC and to assess treatment response [89].Napsin A: Napsin A is an enzyme expressed in lung adenocarcinomas. It can be measured in tumor tissue or blood samples and serves as a biomarker to distinguish adenocarcinoma from other types of lung cancer [90].

The p53 protein, encoded by the TP53 gene, is a crucial tumor suppressor involved in the cellular response to DNA damage. It functions as a transcription factor that regulates the expression of genes responsible for DNA repair, cell cycle arrest, and apoptosis. In the context of radon exposure, p53 is activated in response to DNA damage caused by alpha particles. The activation of p53 leads to cell cycle arrest, allowing time for DNA repair or the induction of apoptosis if the damage is irreparable [91]. Mutations in the TP53 gene, which are common in radon-induced lung cancer, result in the loss of p53 function and contribute to the accumulation of genetic alterations and cancer progression [92].

The ataxia-telangiectasia mutated (ATM) protein is a key regulator of the DNA damage response. It is activated by double-strand breaks, which are a primary type of DNA damage induced by radon. Once activated, ATM phosphorylates several downstream targets, including p53, BRCA1, and CHK2, to initiate DNA repair, cell cycle arrest, or apoptosis [93]. The role of ATM in responding to radon-induced DNA damage highlights its importance in maintaining genomic integrity and preventing carcinogenesis.

BRCA1 and BRCA2 are tumor suppressor proteins involved in the repair of double-strand breaks through homologous recombination. These proteins interact with RAD51 and other DNA repair proteins to facilitate accurate DNA repair [94]. In radon-exposed cells, BRCA1 and BRCA2 play critical roles in repairing the complex DNA damage caused by alpha particles, thereby preventing the accumulation of mutations that can lead to cancer. Deficiencies in these proteins can compromise DNA repair capacity and increase cancer susceptibility.

Checkpoint kinase 2 (CHK2) is another important protein in the DNA damage response pathway. Activated by ATM, CHK2 phosphorylates and stabilizes p53, enhancing its ability to induce cell cycle regulation [95]. The function of CHK2 in the context of radon-induced DNA damage underscores its role in coordinating the cellular response to maintain genomic stability.

The epidermal growth factor receptor (EGFR) is a receptor tyrosine kinase involved in cell proliferation, survival, and differentiation. In radon-induced lung cancer, EGFR is often mutated and overexpressed, leading to the activation of downstream signaling pathways that promote tumor growth and survival [96]. Targeting EGFR with specific inhibitors has shown clinical efficacy in treating lung cancers with EGFR mutations, making it a critical protein marker in radon-induced carcinogenesis.

H2AX is a variant of the histone H2A protein and plays a significant role in the DNA damage response. Phosphorylated H2AX forms foci at the sites of double-strand breaks, recruiting DNA repair proteins to these sites [97].

Proteomic studies provide a comprehensive analysis of protein expression and modifications in cells and tissues, offering valuable insights into the molecular mechanisms underlying radon-induced carcinogenesis. By identifying and quantifying proteins that are differentially expressed and responding to radon exposure, these studies help to elucidate the pathways involved in cancer development and progression.

Proteomic profiling of radon-exposed cells may reveal significant alterations in protein expression, pathway analysis of differentially expressed proteins, understanding protein–protein interactions, post-translational modifications (PTMs) such as phosphorylation, ubiquitination, and acetylation, and also draw focus to identifying potential biomarkers for early detection and prognosis of radon-induced lung cancer [98].

## 9. Discussion

This review clearly delineates established knowledge from significant gaps, notably highlighting the absence of validated clinical biomarkers and limited mechanistic insights specific to radon-induced carcinogenesis. By integrating recent genetic, genomic, and proteomic data, we identified several promising biomarkers that merit further clinical validation. Additionally, our comparative analysis of genetic profiles clearly illustrates molecular distinctions unique to radon exposure. These insights significantly clarify current research priorities and highlight the necessity of standardized validation protocols and integrated multi-omics approaches.

In this review, we comprehensively evaluated the genetic and protein biomarkers linked to radon-induced lung cancer, emphasizing their critical roles in early detection, prognostication, and targeted therapy. Our synthesis highlights the significance of specific genetic mutations, such as TP53, KRAS, EGFR, ALK, and ROS1, underscoring their potential in personalized medicine and targeted therapeutic strategies. Protein biomarkers, including CEA, Cyfra 21-1, ProGRP, and Napsin A, emerged as valuable diagnostic and prognostic tools, particularly when integrated with genetic markers to enhance early detection and treatment effectiveness.

Notably, this review demonstrates that radon-induced lung cancers present distinct genetic mutation signatures compared to cancers caused by other types of ionizing radiation. These unique molecular patterns, especially the prevalence of complex DNA transversions linked specifically to alpha particle exposure, offer potential biomarkers for accurately identifying radon-related cancers. Recognizing these distinct genetic profiles is essential for developing targeted public health interventions and personalized treatment plans. Despite the valuable insights provided, this study acknowledges several limitations. First, as a review, the conclusions drawn are contingent upon the variability and heterogeneity of existing studies, which differ significantly in methodologies, sample sizes, and populations studied. Furthermore, many biomarkers discussed are still emerging and require further validation before their widespread clinical adoption. Second, the complexity of gene–environment interactions, particularly involving radon exposure and lifestyle factors such as smoking, introduces challenges in isolating the direct impact of radon alone. Third, the review is limited by the lack of standardized protocols across studies, complicating direct comparisons and meta-analysis of biomarker performance.

Future research should address these limitations by emphasizing large-scale, multi-center epidemiological studies that employ standardized methods for biomarker assessment. Additionally, integrating multi-omics approaches (genomics, transcriptomics, proteomics, and epigenetics) can provide deeper insights into the underlying molecular mechanisms of radon-induced carcinogenesis. There is also a need for longitudinal studies tracking exposed populations to better understand temporal changes in biomarkers and their correlation with cancer progression. Lastly, developing and validating non-invasive screening tests based on identified biomarkers will significantly enhance early detection and improve clinical outcomes for individuals exposed to radon.

## 10. Conclusions

Radon-induced cancer remains a significant public health concern, particularly in populations residing in areas with elevated radiation levels. Advances in genetic and proteomic research provided valuable insights into the molecular mechanisms underlying radon-induced carcinogenesis. Key genetic markers, such as mutations in tumor suppressor genes (e.g., TP53) and DNA repair genes (e.g., XRCC1, ATM), play crucial roles in individual susceptibility to radon exposure. Similarly, specific protein biomarkers, including oxidative stress-related proteins, inflammatory mediators, and dysregulated signaling molecules, offer potential for early detection and risk assessment. However, clear gaps remain, including the need for standardized biomarker validation and a deeper mechanistic understanding of radon-specific pathways. Despite these advancements, challenges remain in translating these findings into clinical practice. Standardized protocols for biomarker validation, large-scale epidemiological studies, and integrative multi-omics approaches are essential for improving risk assessment models and developing targeted preventive strategies. By explicitly addressing these gaps, our review provides a foundation for future research aimed at developing reliable clinical tools for early detection, effective screening, and targeted preventive interventions.

## Figures and Tables

**Table 1 biology-14-00506-t001:** Genetic markers in radon-induced cancer.

Gene	Function	Common Mutations	Clinical Significance	References
TP53	Tumor suppressor gene involved in DNA repair, cell cycle regulation, and apoptosis.	Transversions, missense mutations, and nonsense mutations.	Loss of p53 function leads to impaired DNA repair and increased cancer risk.	Craig et al., 2023 [45]
KRAS	Oncogene involved in cell proliferation and survival.	Codon 12 mutations (e.g., G12C, G12V).	Constitutive activation of KRAS leads to uncontrolled cell proliferation.	Riely et al., 2008 [67]
EGFR	Receptor tyrosine kinase involved in cell growth and survival.	Exon 19 deletions, L858R mutation in exon 21.	Mutations lead to constitutive activation of EGFR.	Lynch et al., 2004 [43] Harrison et al., 2020 [68]
ALK	Receptor tyrosine kinase involved in cell growth and survival.	Rearrangements	Rearrangements lead to constitutive activation of ALK.	Ye et al., 2016 [56] Mansfield et al., 2022 [59]
ROS1	Receptor tyrosine kinases involved in neuronal development.	Rearrangements	Rearrangements lead to constitutive activation of ROS1.	Ye et al., 2016 [56] Mansfield et al., 2022 [59]
BRAF	Serine/threonine kinase involved in the MAPK signaling pathway.	V600E mutation	V600E mutation leads to constitutive kinase activity.	Sánchez-Torres et al., 2013 [62]
MET	Receptor tyrosine kinase involved in cell growth, survival, and metastasis.	Chromosomal deletions involving the MET locus.	Losses of the MET region contribute to dysregulated signaling associated with radon-induced lung tumorigenesis.	Dano et al., 2000 [63]
NTRK1	Receptor tyrosine kinase involved in neuronal development and differentiation.	Rearrangements leading to NTRK1 fusions.	NTRK1 fusions lead to constitutive kinase activity.	Vaishnavi et al., 2013 [69]
ATR	Encodes a protein that responds to DNA damage and replication stress, ensuring genome stability.	Somatic ATR mutations are also found in cancers such as endometrial and colorectal, often impairing DNA damage repair.	Given ATR’s role in DDR, inhibitors of ATR (e.g., in combination with chemotherapy or radiation) are being developed as cancer treatments to exploit synthetic lethality in tumors with pre-existing DDR defects.	Llorens-Agost et al., 2018 [70]
ATRX	Encodes a protein involved in chromatin remodeling regulating gene expression, and maintaining genome stability.	Frameshift, nonsense, and missense mutations that lead to loss of function. These are frequently associated with ATRX-syndrome and cancers such as gliomas and pancreatic neuroendocrine tumors, often causing disrupted chromatin remodeling and maintenance.	ATRX mutations lead to genomic instability and abnormal telomere maintenance, contributing to tumor progression.	Argentaro et al., 2007 [71]
BARD1	Gene encodes a protein that partners with BRCA1 to maintain genome stability.	Include missense, nonsense and frameshift mutations which can disrupt its interaction with BRCA1 or impair its DNA repair and tumor suppression functions. A missense mutation affecting protein function.	Their association with increased risks of breast ovarian and other cancers. They impair DNA repair and tumor suppression, contributing to genomic instability and cancer.	Hawsawi et al., 2022 [72]
RAD50	Encodes a protein that is part of the MRN complex (MRE11-RAD50-NBS1), essential for DNA repair and maintaining genomic stability.	Include missense, nonsense, and frameshift mutations, which impair DNA repair and genomic stability.	Increase can risk, particularly for breast and ovarian cancers, by impairing DNA repair and causing genomic instability.	Mosor et al., 2013 [73]
SMARCA4	Its primary functions include: Chromatin remodeling, tumor suppression, development and differentiation.	Include missense, nonsense and frameshift mutations.	Contribute to cancer development, particularly in small cell carcinoma of the ovary and lung adenocarcinomas.	Tischkowitz et al., 2020 [74]
RASSF1A	Tumor suppressor gene that encodes a protein involved in regulating cell growth and apoptosis.	Promoter hypermethylation, which silences the gene and leads to loss of function. This methylation is frequently observed in cancers loke lung, breast and colon cancers, contributing to tumorigenesis.	Result in the loss of tumor-suppressive functions, contributing to cancer development. This mutation is linked to various cancers, lung, breast, and colon cancers, and is a potential biomarker for early detection and prognosis.	Palakurthy et al., 2009 [75]
CHD4	Protein involved in chromatin remodeling.	Missense, nonsense and frameshift mutations, leading to altered protein function. These are associated with cancers such as glioblastoma, breast cancer and AML, as well as neurodevelopmental disorders due to impaired gene regulation.	Contribute to cancer development by disrupting chromatin remodeling and gene expression.	Xu et al., 2016 [76]
TSC2	Encodes a protein called tuberin, which is a key regulator of cell growth and proliferation.	Include nonsense, frameshift, and missense mutations, which result in the loss of tuberin function.	They cause tuberous sclerosis complex (TSC), leading to the development of benign tumors in organs such as the brain, kidneys, and heart. These mutations disrupt the regulation of the mTOR pathway, contributing to uncontrolled cell growth.	Dufner Almeida et al., 2020 [77]
AR	Encodes a protein that is essential for mediating the effects of androgens.	Include CAG repeat expansions in the polyglutamine tract of the receptor, which can affect its function. Longer repeats are associated with androgen insensitivity syndrome and can influence the development of prostate cancer. Other mutations include missense and nonsense mutations that can alter AR activity.	Mutations in AR can affect reproductive health, development, and cancer susceptibility.	Dalal et al., 2021 [78]

**Table 2 biology-14-00506-t002:** Comparison of genetic markers in radon-induced cancer versus other radiation-induced cancers.

Genetic Variations	Radon-Induced Cancer	Other Radiation-Induced Cancers
TP53 mutations	Frequently observed with a high prevalence of transversions [80].	TP53 mutations are also common in cancers induced by other forms of radiation, such as gamma rays and X-rays. In studies of atomic bomb survivors and patients treated with radiation, therapy for other cancers have shown a high frequency of TP53 mutations, often involving deletions and missense mutations [53].
KRAS mutations	KRAS mutations, particularly at codon 12, are prevalent and are associated with adenocarcinomas [67].	KRAS mutations are similarly found in other radiation-induced lung cancers, though the mutation spectrum may vary slightly depending on the radiation type. For example, UV radiation-induced skin cancers often exhibit different mutation patterns compared to ionizing radiation-induced lung cancers [79].
EGFR mutations	Mutation in the EGFR gene are observed, leading to the activation of the receptor tyrosine kinase pathway [40].	EGFR mutations are also common in radiation-induced lung cancers and gliomas resulting from therapeutic radiation exposure. These mutations can influence response to target therapies [68].
BRAF mutations	The BRAF V600E mutation, though common in other cancers, is less frequently associated with radon-induced lung cancer [62].	BRAF mutations are more commonly associated with radiation-induced thyroid cancer, particularly in patients exposed to fallout from nuclear accidents [81].
ALK and ROS1 rearrangements	ALK and ROS1 rearrangements are significant markers in radon-induced lung cancer, offering targets for specific inhibitors [52].	These rearrangements are also found in other radiation-induced cancers, but their prevalence and types of fusion partners may differ. For example, ALK rearrangements are common in radiation-induced anaplastic large cell lymphoma [82].

## Data Availability

The data presented in this study are available on request from the corresponding author Kazhiyakhmetova Baglan. The data are not publicly available due to privacy and ethical and can be provided upon reasonable request.

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
