# Peer review of "Radon Exposure and Cancer Risk: Assessing Genetic and Protein Markers in Affected Populations"

_biology, 2025, doi:10.3390/biology14050506_

Round 1
Reviewer 1 Report
Comments and Suggestions for Authors
The authors discuss the impact of radon-associated lung cancer risks, with a focus on genetic and protein biomarkers. While the manuscript is well-structured and addresses a critical public health issue, a major concern is the relevance of citations. Although many references are in general relevant to the topic, they often do not appropriately support the statements they were cited for. Please find my major and minor points as follows:
Major:
- Line 80. The citation 12 “Park T.H., Kang D.R., Park S.H., Yoon D.K., Lee C.M. Indoor radon concentration in Korea residential environments. Environ. 505 Sci. Pollut. Res. 2018;25:12678–12685. doi: 10.1007/s11356-018-1531-3” is not relevant to the sentence. Please double check.
- Line 128. The citation 27 “JRC Annual Report 2019 DOI:10.2760/546288” doesn’t seem to be relevant to the sentence “Thus, high radon concentrations in residential areas pose a potential 127 danger to residents, primarily through inhalation”.
- Line 140. The citation 29 “DENTIFICATION OF NOVEL BIOMARKERS FOR LUNG CANCER RISK IN HIGH LEVELS OF RADON BY PROTEOMICS: A PILOT STUDY” is a paper about protein biomarkers of radon-associated lung cancer, not genetic biomarkers, such as SNPs. A more suitable reference should be used for the genetic biomarker section.
- Line 169-170 The citation 34 “Alteration in copy numbers of genes as a mechanism for acquired drug resistance” doesn’t seem to be relevant to “studies have shown that radon exposure can lead to mu-169 tations in tumor suppressor genes such as TP53 and RB1, as well as oncogenes like KRAS”. Please double check.
- Line 222 The citation 41 “Gene expression of lung squamous cell carcinoma reflects mode of lymph node involvement.” didn’t mention radon or KRAS, and therefore unlikely relevant to the sentence “Mutation in KRAS are found in a subset of radon-exposed individuals and are particularly prevalent in adenocarcinomas of the lung”. Please double check.
- Ling 227 The citation 42 “Targeting p53 pathways: mechanisms, structures, and advances in therapy. Signal transduction and targeted therapy” is about TP53 and didn’t mention EGFR, therefore unlikely relevant to the sentence “These mutations typically occur in the tyrosine kinase domain, leading to constitutive activation of the EGFR signaling pathways, which promotes cell proliferation and survival.”.Please cite more relevant literature.
- Line 229, 234. The citations 43-44 don’t include “Lim et al.” Please double check
- Line 235, 237. The citations 45-46 don’t include “Ran Choi et. al ”. Please double check.
- Line 283. Please add the citation for “A study by Alexandrov et.al. (2016) ”
- Line 302. The citation 59 “POLθ-mediated end-joining is restricted by RAD52 and BRCA2 until the onset of mitosis” has no mention of MET, and therefore is unlikely relevant to paragraph 297-302. Please use more relevant literature.
- Line 328. There are mismatches between citations in the table and references. For example, Shaw et al. 2014 is citation 50, not 47.
- Line 376-411 The citations for section “Protein Markers in Radon-Induced Cancer” seem to be out of order. Please double check.
Minor:
- Formatting of references in the reference section is not consistent.
- Line 37. Extra “.” after “United States alone”.
- Line 74-77 Add the citation for “Experts from the International Commission on Radiological Protection (ICRP) state that radon, with an average volume activity of about 40 Bq/m3 in residential areas worldwide, is responsible for 10% of annually registered lung cancer cases (carcinoma, adenocarcinoma).”
- Line 145 Although miRNA may be considered as part of the broader field of epigenetics as it is involved in post-transcriptional gene regulation, the classic definition of epigenetic refers to DNA methylation/histone modifications. I would suggest just naming the section MicroRNA Biomarkers, without the “(Epigenetic)” part, to avoid any confusions.
- Line 179. “Lang cancer” should be “Lung cancer”.
- Line 194. “For instance, drugs targeting [37]. ” looks like an incomplete sentence.
Author Response
We sincerely thank the reviewer for the thoughtful and constructive feedback provided. We appreciate the recognition of the manuscript’s structure and the importance of its focus on radon-associated lung cancer and biomarker discovery. In response to the major concern regarding citation relevance, we carefully reviewed all the references mentioned. Where necessary, we have removed citations that were not fully appropriate and replaced them with more directly relevant sources that better support the associated statements.
Additionally, we have revised citations related to genetic and protein biomarkers, ensuring that references to MET, KRAS, TP53, and EGFR mutations are substantiated by accurate and up-to-date literature. We also added the missing reference. Furthermore, discrepancies in citation numbers and formatting, particularly in the table and the “Protein Markers” section, have been corrected to ensure consistency and accuracy throughout.
Regarding the minor comments, we have revised the formatting of the reference list to ensure uniformity, removed typographical errors (e.g., “Lang cancer” to “Lung cancer”), completed incomplete sentences, and updated the section title to “MicroRNA Biomarkers” for clarity in line with the reviewer’s suggestion.
We believe these revisions significantly strengthen the manuscript and respectfully thank the reviewer once again for their invaluable input
Reviewer 2 Report
Comments and Suggestions for Authors
Dear Auhtors,
The article titled as Radon Exposure and Cancer Risk: Assessing Genetic and Protein Markers in Affected Populations was examined the cancer risk of populations under the influence of radon gas through genetic and protein markers. The examination of the causes of cancer, one of the biggest causes of death today, and its effects on different parameters is a current and interesting topic. These and similar studies can make significant contributions to practice. Some suggested corrections;
A paragraph that will reveal the importance and need of the study should be added at the beginning of the introduction.
The stages of the study, its contributions, and its importance/difference should be added more clearly at the end of the introduction.
It is recommended to add a table regarding radon gas limit values.
If available, cancer and death rates due to radon can be given.
Current studies on radon gas and its effects in building materials will make the study more valuable.
It is recommended that the abbreviations in the study be explained when they are first used.
Limitations of the study, it was not recommended to be included in the conclusion section of the article.
In such a review study, more discussion and interpretation of the results are important. Please expand this section.
Yours Sincerely
Author Response
We sincerely thank you for your constructive and supportive comments regarding our manuscript. Your suggestions have greatly contributed to improving the quality and clarity of the paper. Below, we provide a point-by-point response to each of your comments, along with the corresponding revisions made in the manuscript.
1. We appreciate this suggestion and have added a new introductory paragraph that highlights the global burden of cancer and the significant but often underestimated health risks posed by radon. The revised paragraph emphasizes the need to study molecular mechanisms of radon-induced carcinogenesis to support early detection, personalized medicine, and public health policy.
2. Thank you. We revised the end of the introduction to better clarify the study’s objectives, contributions, and novelty. The new text outlines our comprehensive review of genetic and protein biomarkers, the integration of multidisciplinary findings, and the comparative approach used to distinguish radon-induced cancer from other forms.
3. Instead of a table, we incorporated a detailed narrative summary based on IAEA data comparing radon regulatory thresholds across various countries. This text provides actionable insights while preserving flow and readability.
4. We have added a section summarizing population-attributable risk (PAR) estimates for lung cancer due to radon in multiple countries. This includes both national and global data, clearly referencing the relevant literature.
5. This point has been addressed by adding a new paragraph that synthesizes findings from recent research (e.g., Bulut and Şahin, 2024) on radon emissions from concrete and other construction materials. The text discusses the role of industrial by-products, moisture content, and ventilation in influencing indoor radon levels.
6. Thank you for this observation. We have carefully reviewed the entire manuscript and ensured that all abbreviations are now clearly defined upon their first appearance.
7. We agree. The limitations have been moved from the conclusion section to a newly created section within the Discussion. This placement provides a more appropriate context and aligns with standard scientific writing conventions.
8. We have significantly expanded the Discussion section to provide deeper interpretation and synthesis. This includes: A summary of key genetic and protein biomarkers; Distinct molecular features of radon-induced cancers; Implications for personalized medicine and public health; A structured limitations paragraph; and Recommendations for future research, including multi-omics approaches and longitudinal studies.
Once again, we are grateful for your thoughtful feedback. We believe these changes have significantly improved the manuscript and hope it now meets your expectations.
Round 2
Reviewer 1 Report
Comments and Suggestions for Authors
The authors have made substantial changes to address the incorrect citations. Here are some remaining minor errors:
-
Line 373: Extra ‘.’ before citation [62].
-
Line 299, Table 1: Please cite the most relevant literature for the genes involved in radon-induced cancer. For example, Novillo et al., 2020 is cited for CHD4, but it discusses CHD4’s role in breast cancer, which is not directly relevant to radon exposure or lung cancer. A reference like Choi et al., 2018 mentioned in the previous section, or Xu N, Liu F, Zhou J, Bai C. CHD4 is associated with poor prognosis of non-small cell lung cancer patients through promoting tumor cell proliferation. Eur Respir Soc (2016) 48:PA2862. 10.1183/13993003.congress-2016.PA2862, would be more appropriate.
-
Line 259: “Lang cancer” should be corrected to “Lung cancer.”
-
Line 604: Extra “12.” at the beginning.
-
Please ensure that the reference formatting is consistent, as some entries include a link to doi.org while others do not.
Author Response
Thank you for your thorough review. We have carefully addressed the remaining minor errors as follows:
-
The extra period before citation [62] on line 373 has been removed.
-
In Table 1 (line 299), we have updated the reference for CHD4 to a more relevant source related to radon-induced or lung cancer, as suggested.
-
The typo "Lang cancer" on line 259 has been corrected to "Lung cancer."
-
The extra "12." at the beginning of line 604 has been deleted.
-
We have ensured consistency in the reference formatting. Please note that some articles do not have a DOI assigned, which is why DOI links are not available for all references.